# Method Development and Qualification of pH-Based CEX UPLC Method for Monoclonal Antibodies

**DOI:** 10.3390/biotech11020019

**Published:** 2022-06-03

**Authors:** Mithun Bhatt, Anshu Alok, Bhushan B. Kulkarni

**Affiliations:** 1Mehsana Urban Institute of Sciences, Faculty of Science, Ganpat University, Mehsana 384012, India; biotech.mit@gmail.com; 2Department of Biotechnology, UIET, Panjab University, Chandigarh 160014, India; anshualok2@gmail.com; 3Dr. Prabhakar Kore Basic Science Research Centre, K.L.E. Academy of Higher Education and Research, Belagavi 590010, India

**Keywords:** mAbs, charge variant analysis, “PTMs”, ion exchange chromatography, mass spectrometry, biosimilar, pH gradient, quality control

## Abstract

Post-translational modifications (“PTMs”) in monoclonal antibodies (mAbs) contribute to charge variant distribution, which will affect biological efficacy and safety. For the characterization of mAbs, charge variants are used as a critical quality attributes for product quality, stability consistency and effectiveness. Charge variants in mAbs are characterized by a time-consuming and a multistep process starting from cation/anion exchange chromatography, acidic/basic fractions collection and subsequent reverse phase (RP) liquid chromatography, coupled with mass spectrometry (MS) analysis. Hence, an alternative characterization approach that would be highly selective for ion exchange chromatography-based charge variant analysis, which is compatible with on-line MS detection, is needed in the biopharma industry. Against this backdrop, multiple studies are being conducted to develop a simple straight on-line charge variant analysis method. In this regard, we apply the current study, which aims to develop a charge variant analytical method, based on volatile buffers with low ionic strength that can be used for on-line MS detection of charge variants of mAbs. This would enable the detection on “PTMs” using low ionic strength mobile phase compatible with MS. Hence, fruitful data can be obtained with a single chromatography run without any test sample preparation, eliminating the need for multiple steps of analysis, time-consuming process and multiple sample preparation steps. Thus, Charge Variant Analysis-MS technique will allow the characterization of charge-related PTMs on the intact protein stage. In this regard, this study is about development of a method having combination of chromatography and volatile mobile phase for mass spectrometry detection of mAbs being analyzed in native form. The method is qualified considering pharmacopeia guidelines because the ultimate aim is to transfer this method for Quality Control (QC) release testing of a monoclonal antibody, which is critical for batch release and the regulatory point of view. Acidic and basic variants have been separated with high resolution peak profile. Furthermore, there was no matrix interference and good separation selectivity in terms of specificity was obtained using this method. The experimental data suggested for the linearity of the method are 2.4 mg/mL to 3.6 mg/mL with % RSD below 2.0%. Additionally, Limit of Quantitation is found to be 0.15 mg/mL, which is 5% of loading amount. Consistently, the data show that the method is precise under the same operating conditions with a short time interval. Overall a simple, accurate, robust and precise pH gradient cation exchange chromatography method was developed and qualified for the characterization of a therapeutic native mAb. Additionally, this method can be used to claim a biosimilar product profile of an in-house product compare to an innovator.

## 1. Introduction

In recent times, mAbs-based therapeutics products are rapidly moving up in biosimilar industries [1]. It is well-known that mAbs have great clinical applications for the treatment of immunological disorders, malignancy and other critical infections. To obtain approval for clinical use as a biosimilar product, bio similarity data must be generated with an innovator molecule with various analytical tools. The isoelectric point of a protein is the pH, at which a protein carries no net charge and hence is considered neutral. The isoelectric point (pI), where the net charge of the protein is neutral, is pH-dependent. The pI of a protein is crucial to understanding its biochemical function [2]. The pI of mAbs is mostly dependent on the dissociation constant of the ionizable groups. Amino acids containing acidic or basic functionalities form the peptide bounds, which ultimately give rise to the polypeptides or the proteins. These functionalities influence the overall surface charge, which, in turn, can be regulated by monitoring the pH of buffer system surrounding the protein.

Quality by design (QbD) is a next generation step in developing and producing biopharmaceutical products in the biopharma industry. The best quality and most consistent product can be generated using the QbD approach for the characterization of therapeutic proteins. [3]. mAbs and related products have modifications such as aggregation, deamidation, disulfide bond scrambling, fragmentation, glycosylation, lysine truncation, oxidation, isomerization and pyro glutamate formation. These modifications may impact the critical quality attributes (CQAs) of the drug with relation to its efficacy and safety. The composition and physiochemical parameters of the cell culture media at the time of up-stream fermentation can be impacted by critical process parameters [3,4].

Therapeutic mAbs usually display heterogeneity when they are produced using cell culture methods. These heterogeneities or variations are caused by deamidation, glycosylation, mutation, N-terminal glutamine cyclization, C-terminal lysine truncation, and oxidation, resulting in isoforms of the peptides. Due to the aforementioned causes, mAbs tend to display hurdles with respect to their characterization, of which charge variants are an important attribute. Charge deviations are natural features, not impurities of this type of therapeutic mAbs, and they have critical impacts on protein stability and bioactivity.

mAbs and other therapeutic proteins are routinely characterized, processed and analyzed for charge heterogeneity by ion exchange chromatography (IEX), as this is a gold standard technique for assessing the quality of proteins and to allow for separate control of individual proteoforms. To assess the charge heterogeneity, IEX relies on electrostatic interactions between the resin functional groups (cation/anion) and mAbs. The purification and characterization of charge sensitive mAbs can be carried out using cation exchange chromatography (CEX) which is based on the pI of the mAbs. However, due to the minute differences in the binding properties charge variant mAbs, separation sometimes becomes difficult using CEX. Tiny particle analytical stationary phases called column and mobile phase gradient chromatography are used for separation.

In biopharmaceutical industry, charge variant analysis is a critical attribute used for direct comparison of biosimilar product lots with reference products to certify constant product quality, which is very important for biosimilar product profile.

Currently, in the biopharmaceutical industry, the tool of choice for quality control, optimization and product retrieval from upstream and downstream processing is analytical technology [5]. However, for identification of the unknown peaks in charge variants analysis which are out-of-specification for product manufacturing, it requires identification and characterization, which is a time-consuming process, involving initial sample preparation procedures and follow-up experiments. These experimental procedures generally involve peak collection followed by buffer exchange to carry out further analysis [6,7,8]. The entire identification and characterization process leads to more time and more cost. Sometimes, these experimental protocols lead to forceful generation of modifications in the protein product that are actually not present in its native form [9,10].

mAbs are traditionally eluted in IEX using salt gradients of low-to-high concentration. This eluent, however, is not suitable for MS analysis, due to its high salt concentration. Furthermore, due to denaturation of tertiary structure and exposure of charged residues, highly charged mAb ions are produced, which results in the generation of complex MS [11].

Traditional ion exchange chromatography uses salt gradients of low-to-high concentration to elute proteins/mAbs, but when connecting to mass, the high concentrations of salt can be problematic, which is not recommended. Another limitation of this approach is denaturation of the protein/mAbs, resulting in the generation of complex mass spectra containing highly charged mAb ions due to the loss of tertiary structure and exposure of charged residues [12,13,14,15].

The ICH Q2 (R1) guideline, which is directly related to the validation of analytical methods for conventional medications but can also be utilized with biopharmaceutical products, should be followed when qualifying analytical processes employed in the characterization of biopharmaceuticals products. The method qualification of a stability-indicating parameter requires the analysis of stressed or force-deteriorated samples to show that the method is suitable for active ingredient analysis in the presence of the degradation conditions. The method can be submitted to the appropriate regulatory agency for approval as part of the product license application if the qualification findings are satisfactory and within the limitations of ICH recommendations. The goal of this research is to create and validate a simple, precise, and robust pH gradient-based IEX approach for determining charge variations in mAb samples that can be replicated in any lab. This method can be used in routine testing for quality control analysis, giving manufacturer’s confidence in the quality of their products. Furthermore, the approach can be used to track the stability of mAb in biopharmaceutical formulations under a variety of force-degraded circumstances. For regulatory data submission of biosimilar products, the analytical method must be qualified and validated for analysis. Thereby, in this study, the method is qualified as per ICH guidelines for quality control batch release use and regulated data submission.

Additionally, the aim of this work is to develop and qualify a mass spectrometry-based analytical method for charge variant analysis based on volatile pH-based buffers with low ionic strength. This allows for on-line mass spectrometric detection of charge variants in different mAbs/Proteins. Low ionic strength eluents, which are employed for MS detection and have a low buffering capacity, are a feature and quality of this approach. A sufficient amount of data can be acquired from a single chromatographic injection without the need for any sample preparation operations, which would ordinarily necessitate many phases of analysis and various sample preparation procedures [13,14]. As a result, the charge variant analysis-MS technique will enable the monitoring of many quality parameters at the intact protein level, which will aid in process optimization both upstream and downstream. It is also possible to investigate new information on protein heterogeneity and breakdown ability, which may have implications for the launch of prospective biosimilar products in the future.

## 2. Materials and Methods

### 2.1. Materials

Ammonium bicarbonate, water (mass grade, JT Baker, Phillipsburg, NJ, USA), ammonium hydroxide solution, and acetic acid (ACS mass grade) were used for analysis. MAbs (Bevacizumab) was kindly provided from the university. It has a molecular weight of about 149 kDa and pI around 8.3. Bevacizumab is a humanized monoclonal IgG1 antibody, and inhibits angiogenesis by binding and neutralizing VEGF-A. The concentration of IgG1 is claimed as 25 mg/mL.

### 2.2. Method

CEX-UPLC analyses of untreated Bevacizumab was carried out on an UPLC system (Binary, Waters). Separation was performed on a MAbPac SCX-10 RS 5 µm, 2.1 × 50 mm column (Thermo Fisher Scientific, Sunnyvale, CA, USA) using a mobile phase A comprising 25 mmol·L^−1^ ammonium bicarbonate and 30 mmol·L^−1^ acetic acid in water (pH 5.3) and a mobile phase B comprising 10 mmol·L^−1^ ammonium hydroxide in water (pH 10.9). A pH gradient was applied consisting of 35 to 60% mobile-phase B in 7 min followed by a flushing step with 90% B and a re-equilibration step with 35% B for 5 min, flowrate 400 µL min^−1^, T 25 °C, and UV detection at 280 nm. Per run, 30 µg of IgG1 antibody was injected.

Result: Based on Table 1, the system parameters program, CEX UPLC run was performed using Bevacizumab IgG1 antibody sample with a 30 microgram injection volume on the column. Total run time was 20 min. The representative chromatogram is shown as below.

As per Figure 1 chromatogram, main peak which is obtained around 7 min represent IgG1 purity having charge variants as pre peak (acidic variants) and post peak (basic variants). Those charge variants are present due to PTMs. From this it is very clear that the method is producing quite good resolution in acidic and basic variants separation using volatile mobile phase gradient. Additionally, the method run time is short, around 20 min, which shows that the method is compact and provides a platform to characterize charge variant at intact level within a short time period.

### 2.3. Method Qualification

Transfer of analytical qualified method to the quality control lab of mAbs/proteins is a crucial final step in the acceptance of this method, which has quality attributes parameters that are under a good scan. Method qualification is a systematic approach that states with assurance that a specific process will meet its pre-defined acceptance criteria. Additionally, an effective analytical method qualification will improve the precision and accuracy of the previously developed chromatographic methods and decrease their errors. Development of a qualified ion exchange method will help to avoid time-consuming experiments and product developmental and manufacturing cost. In analytical approach, before a developed chromatographic method can be implemented into quality system, it must first be qualified and validated and shown that it is a suitable method for the reporting critical quality data generation. If the qualified analytical method data are acceptable, the method can be send to the appropriate regulatory authority for approval of product license application. Validation of analytical methods used in the characterization of biopharmaceuticals industries should be performed in compliance with the ICH Q2 (R1) guideline, which outlines the validation of analytical procedures. The qualification of a stability-indicating method requires analysis of stressed samples to show that the method is capable of analyzing pure protein in the presence of the degradation of products.

## 3. Qualification Parameters Evaluation

### 3.1. Linearity

The linearity of an analytical method is its ability to obtain test results which are directly proportional to the amount/concentration of analyte in the protein sample within a specified range. Linearity of the CEX method was performed up to 3.6 mg/mL. For this study, mAb RMP Bevacizumab working standard solutions of 2.4, 2.7, 3.0, 3.3 and 3.6 mg/mL were prepared in duplicate from 25 mg/mL and injected into the Waters UPLC instrument. A calibration curve was plotted for main peak area vs. drug concentration/amount and a coefficient of determination (R^2^) is calculated by linear least-square regression mathematical analysis.

Observation: As per chromatogram in Figure 2, calibration curve obtained from Figure 3 and compiled data obtained from Table 2, it can be concluded that CEX UPLC method has load amount linearity from 80 to 120%, with respect to main peak area. It also shows R^2^ 0.99 with % recovery between 95 and 105%, suggesting that the method is linear within 10% of recovery.

### 3.2. Limit of Detection (LOD) and Limit of Quantification (LOQ)

The limit of detection for analytical method is the lowest amount of analyte/protein in a sample which can be detected but not exactly quantitated as a reporting value. The limit of quantitation for analytical procedure is the lowest amount of analyte/protein in a sample which can be quantify with accuracy and precision. The LOQ limit is a parameter of quantitative value for low levels of compounds/impurities in sample matrices. It is used for the determination of impurities for force degradation products. The LOQ or LOD is reported by either signal-to-noise ration or impurity level area determination. For LOD and LOQ determination, Bevacizumab RMP was injected at 5% loading amount, which is 0.15 mg/mL on the column. Back-calculated recovery was determined using slope and intercept.

Observation: Based on above chromatography data obtained from Figure 4, Figure 5 and Figure 6 and data Table 3, it is concluded that LOQ of the method is 5% of loading sample amount out of 100% which is 0.15 mg/mL. This method can detect LOD <5% of loading sample amount but this cannot be quantitated.

### 3.3. Precision and Accuracy

The precision of an analytical method is based on the closeness of agreement (degree of scatter) between a linear series of measurements taken from multiple concentrations of the same sample under the defined conditions. Precision can be determined at three levels: repeatability, reproducibility and intermediate precision.

Repeatability: Repeatability provides information for the precision under the same operating conditions with a short time interval. Repeatability can also be named as intra-assay precision.

Reproducibility: Reproducibility provides the precision between labs.

Intermediate precision: Intermediate precision provides data regarding within-laboratories variations: different days, different analysts, different equipment, different column and different mobile phase preparation variation.

The accuracy of an analytical method is closeness of agreement between the value which is accepted as a true value or an accepted reference value and the value obtained. Accuracy is defined as the closeness of the measured value to the actual value. The determination of accuracy was carried out with calculation of the average recovery value by analyzing three standard solutions triplicates at three concentration levels high (2.00 mg/mL), the medium (0.50 mg/mL), and the low (0.12 mg/mL) of the linear range of the calibration curve.

The precision of the method was performed as the intra-day and inter-day precision, which represent repeatability and intermediate precision, respectively. It is defined with given as relative standard deviations (RSDs). Intra-day precision was determined using 12 samples with same concentration levels of 3.0 mg/mL injected on UPLC system. These samples were prepared and injected into the chromatography system on different days with different system, different column, and different analyst.

Observation: Based on above Figure 7 and Figure 8 and Table 4, Table 5, Table 6 and Table 7 data, it is found that repeatability data % RSD of six different injections for analyst 1, analyst 2 and intermediate precision data for a combined 12 injections with respect to main peak and total peak area is found to be RSD < 5.0%. This data shows that the method can be precise for the purpose of reporting charge variants percentage with precision of 95–105%.

Table 7 accuracy experiment data show that it is falling 50 to 150% of analyte loading amount with <±10% recovery.

### 3.4. Specificity and Stability Indicating Property

Specificity is defined as two major qualification parameters evaluation as Matrix interference and selectivity. It is the ability to assess and measure accurate and specific analyte of interest in the presence of other additives, including impurities, degradants, matrix, etc. It must be demonstrated that our analytical method is not affected by any impurities or excipients. For the specificity experiment, an IgG1 antibody concentration of 3.0 mg/mL injected with buffer blank was used to check whether any buffer peak has interference with the elution time of the Bevacizumab peak.

Observation: As per Figure 9 chromatography overlay data the matrix components did not interfere with the signal obtained for the analyte and thus there was no matrix interference. The method is therefore specific for evaluating the identity of the BevacizumAb sample.

### 3.5. Robustness

The robustness is defined as a measure of a method’s capacity to remain unaffected by small but deliberate changes in procedural parameters. Examples of typical variations are stability of analytical solutions, variations in pH of solvents, column temperature, auto sampler temperature, and flow rate variation. For this study, mAb/protein at a concentration of 2.00 mg/mL was analyzed to evaluate the impact of each modified condition on the data results. Bevacizumab sample was run on UPLC system at initial time point and after 48 h kept in an auto sampler (2–8 °C) to check the sample stability.

Observation: With the Figure 10 robustness data it can be concluded that the mAb sample which is kept in an auto sampler and the same mobile phase can be used for up to 48 h after preparation.

## 4. Discussion

This method can be used for charge-related variant analysis of mAbs/proteins and can be used for on-line mass detection, in terms of characterization process, to address acidic and basic PTMs. Using Bevacizumab as an example, we showed that chromatographic gradient optimization will result in separation and identification of charge-related impurities that are chromatographically resolved on the UPLC system.

We have also shown the method qualification data, which state that the method is effective for use in QC testing for routine analysis for the biopharma industry.

## 5. Conclusions

In this characterization study, a simple, accurate, robust and precise pH gradient CEX chromatography method was developed and qualified for the characterization of a therapeutic native mAb. The results of method qualification on UPLC studies showed that the developed ion exchange method was linear (R^2^ = 0.99) in linearity data with % RSD less than 10% with concentration range (0.3–4.50 mg/mL), in which the method was able to separate charge-related variants of mAb and quantify it with specificity. The accuracy, precision, and robustness of the CEX method meet the criteria of the biopharmaceutical industry therapeutic protein characterization analysis. This method is stability-indicating and informative for Mab/proteins, which can be quantified without interference. As per our knowledge, this is the first study to qualify a CEX method to characterize therapeutic mabs/proteins according to the ICH guideline which can be reproduced in a QC lab. Method qualification is an essential step in introducing IEX methods to quality control laboratories in the biopharma industry for batch release. Therefore, the method can be used for the routine analysis of MAb/proteins, evaluation of stability samples, and quality control of in-process, drug substance, drug product samples of mAb in the biopharma industry.

## Figures and Tables

**Figure 1 biotech-11-00019-f001:**
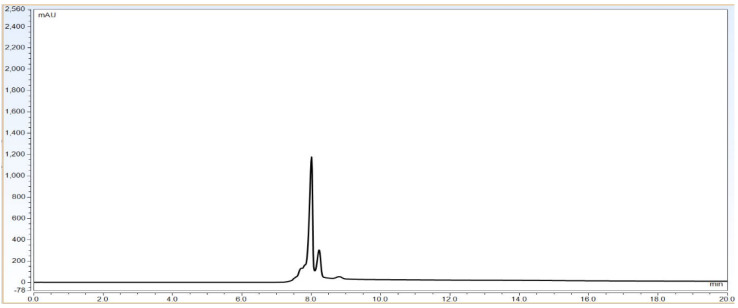
Representative chromatogram of IgG1 antibody using pH based CEX UPLC method. The gradient program and mobile phase information is mentioned in Section 2.

**Figure 2 biotech-11-00019-f002:**
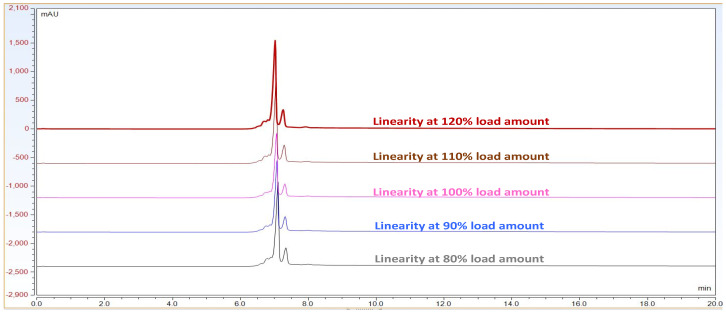
Representative CEX UPLC analysis data stacked overlay chromatogram for load linearity data from load ranging from 80 to 120%. This shows that the method is linear with good resolution in pre and post peaks from 80–120% of sample loading amount on column. *X*-axis represent time in mins. Additionally, *Y*-axis represents signal in mAU.

**Figure 3 biotech-11-00019-f003:**
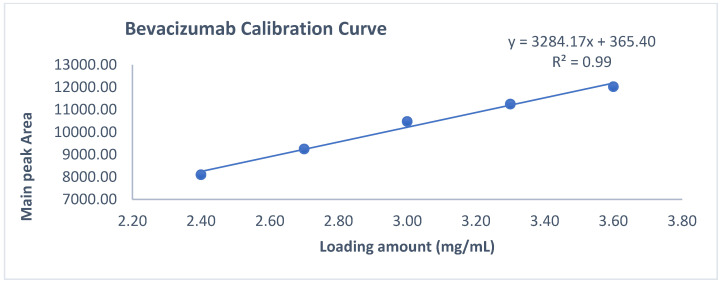
Slope and intercept calculation using loading amount versus main peak area obtained form 80–120% of loading amount.

**Figure 4 biotech-11-00019-f004:**
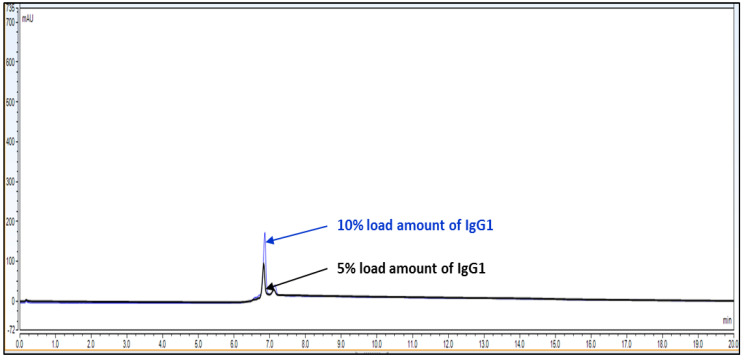
Overlay representative chromatogram for IgG1 sample loading of 10% and 5%, i.e., 0.30 mg/mL and 0.15 mg/mL from 100% standard injection amount, i.e., 3.0 mg/mL for LOQ calculation.

**Figure 5 biotech-11-00019-f005:**
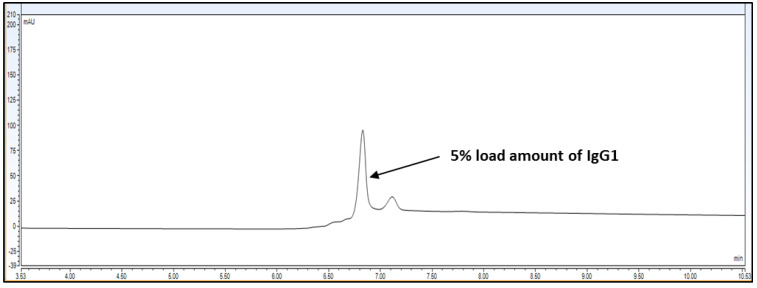
Representative chromatogram for IgG1 sample loading of 5%, i.e., 0.15 mg/mL from 100% standard injection amount, i.e., 3.0 mg/mL for LOD calculation.

**Figure 6 biotech-11-00019-f006:**
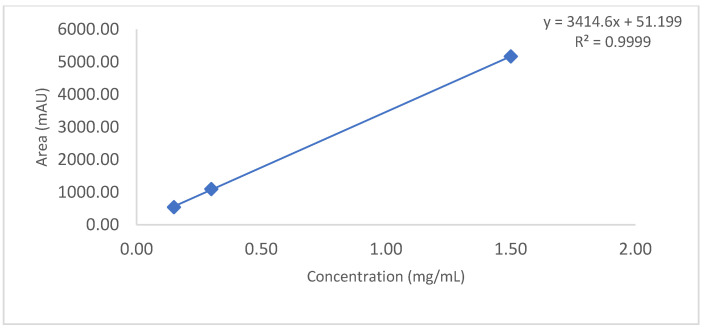
Slope and intercept calculation using loading amount (mg/mL) versus main peak area (mAU) obtained from 1.5 mg/mL to 0.15 mg/mL.

**Figure 7 biotech-11-00019-f007:**
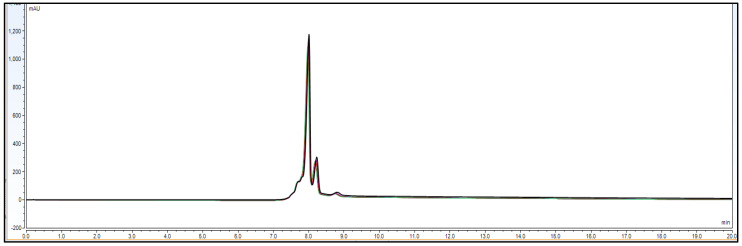
Chromatographic overlay for repeatability experiment using 6 different sample preparations of IgG1 sample 3.0 mg/mL concentration.

**Figure 8 biotech-11-00019-f008:**
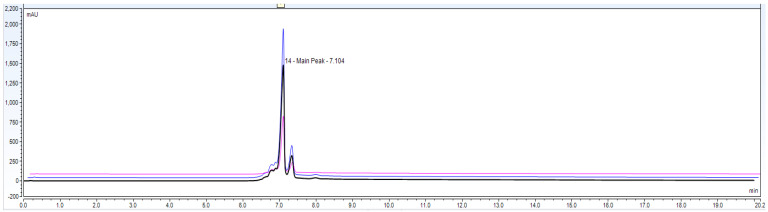
Accuracy data chromatogram with 50, 100 and 150% of loading amount.

**Figure 9 biotech-11-00019-f009:**
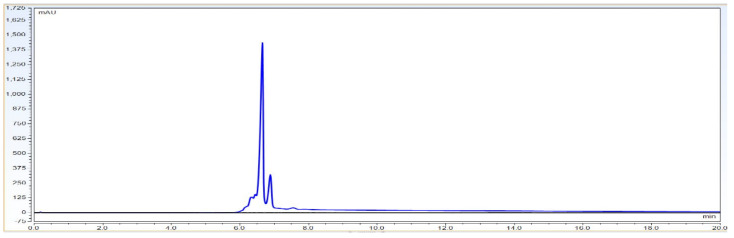
Representative chromatographic overlay data for IgG1 sample and buffer blank run. These data show that the method is specific enough to claim charge variants analysis and that it has no matrix interference at IgG1 peak elution.

**Figure 10 biotech-11-00019-f010:**
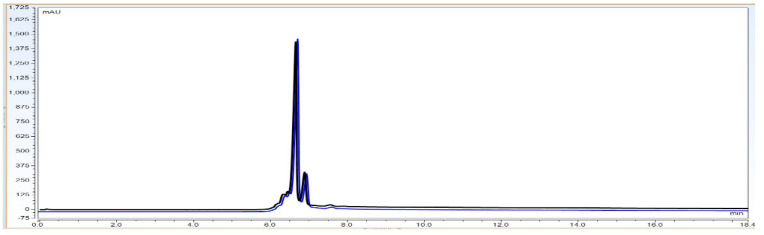
Robustness experiment overlay of Bevacizumab sample (3.0 mg/mL concentration) on CEX UPLC system at 0 time point and 48 h kept in auto sampler with the same mobile phase.

**Table 1 biotech-11-00019-t001:** Representative CEX UPLC method run program with Mobile phase A and B gradient composition. These system parameters are applicable for waters Binary UPLC system.

UPLC Gradient Time (min)	Flow Rate (mL/min)	% B Mobile Phase	UPLC System Curve Value
0	0.4	35	6
3		35	6
10		60	6
12		90	6
12.5		90	6
15		35	6
20		35	6

**Table 2 biotech-11-00019-t002:** Compiled linearity set data in duplicate injection with loading amount of IgG1 sample from 80 to 120%. Slope and intercept are taken from Figure 3.

			Intercept	3284.17		Slope	365.40	
Loading Amount of IgG1 (mg/mL)	Loading Amount %	Main Peak Area (mAu × s) Inj. 1	Main Peak Area (mAu × s) Inj. 2	Average Area (mAu × s)	SD	% RSD	Back Calculation Conc. (mcg)	% Recovery
3.60	120	12,021	12,027	12,024.00	4.24	0.04	3.49	97
3.30	110	11,275	11,221	11,248.00	38.18	0.34	3.19	97
3.00	100	10,460	10,487	10,473.50	19.09	0.18	2.89	96
2.70	90	9228	9261	9244.50	23.33	0.25	2.59	96
2.40	80	8097	8102	8099.50	3.54	0.04	2.29	95

**Table 3 biotech-11-00019-t003:** % Recovery calculation from loading amount of 1.5 mg/mL to 0.15 mg/mL IgG1 sample on column. This data ensure that at 0.15 mg/mL of sample injection % recovery is 90%, which indicates the high sensitivity of the CEX UPLC method.

			Intercept	3414.6000		Slope	51.199	
Loading Amount (mg/mL)	Loading Amount %	Area (mAu × s) Inj. 1	Area (mAu × s) Inj. 2	Average Area (mAu × s)	SD	% RSD	Back Calculation Conc. (mcg)	% Recovery
1.50	50	5077	5264	5170.50	132.23	2.56	1.49	99
0.30	10	1073	1124	1098.50	36.06	3.28	0.29	95
0.15	5	544	542	543.00	1.41	0.26	0.14	90

**Table 4 biotech-11-00019-t004:** Qualification data set for repeatability set 1 experiment using 6 different samples of 3.0 mg/mL concentration.

No. of Preparations	Main Peak Area	Total Area
Prep 1	10,715	14,898
Prep 2	10,662	14,887
Prep 3	10,633	14,891
Prep 4	10,547	14,798
Prep 5	10,509	14,718
Prep 6	10,458	14,640
Average	10,587	14,805
SD	98.44	107.34
% RSD	**0.93**	**0.73**

**Table 5 biotech-11-00019-t005:** Qualification data set for repeatability set 2 experiment using 6 different samples of 3.0 mg/mL concentration.

No. of Preparations	Main Peak Area	Total Area
Prep 1	10,355	15,549
Prep 2	10,360	15,635
Prep 3	10,330	15,595
Prep 4	10,328	15,765
Prep 5	10,436	15,536
Prep 6	10,344	15,326
Average	10,359	15,568
SD	39.93	144.20
% RSD	**0.39**	**0.93**

**Table 6 biotech-11-00019-t006:** Intermediate precision data compilation based on set 1 and set 2 repeatability experiments.

	Main Peak Area	Total Area
TOTAL AVERAGE	10,473	15,187
TOTAL SD	139.17	416.16
TOTAL % RSD	**1.3**	**2.7**

**Table 7 biotech-11-00019-t007:** Recovery calculation for accuracy data set.

			Intercept	5098.2000		Slope	550.5	
Loading Amount (mg/mL)	Loading Amount %	Area (mAu × s) Inj. 1	Area (mAu × s) Inj. 2	Average Area (mAu × s)	SD	% RSD	Back Calculation Conc. (mcg)	% Recovery
4.50	150	22,214	22,612	22,413.00	281.43	1.26	4.61	102
3.00	100	14,696	14,705	14,700.50	6.36	0.04	3.11	104
1.50	50	6989	7248	7118.50	183.14	2.57	1.61	107

## Data Availability

Data will be made available upon reasonable request to corresponding author.

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
