# Peer review of "Method Development and Qualification of pH-Based CEX UPLC Method for Monoclonal Antibodies"

_biotech, 2022, doi:10.3390/biotech11020019_

Round 1
Reviewer 1 Report
This work discusses the development of a CEX based method compatible with MS for detection of charge variants for mAbs. I think the overall concept of the paper has some merit but the authors need to address the following concerns:
- It is not true that when pH/pI > 1, the mAb exhibit a complete positive charge and bind to a cation exchanger. This is a complex topic and the presence of charged patches matter. This is even more true at low conductivities for mAbs. So this statement needs to be reworded carefully for mAbs.
- In the methods section, the authors barely give information on the type of IgG. IgG2? IgG4? What is the pI? How many IgGs were tested? Without this information, this paper is not of much value
- I see what the authors are trying to accomplish -- and it has value -- in terms of reproducibility and robustness, but how is this different from existing published work in this field? I see no mention of other literature that has worked on similar methods (there are authors who have worked on this and published -- fairly common in industry)
Author Response
1. It is not true that when pH/pI > 1, the mAb exhibit a complete positive charge and bind to a cation exchanger. This is a complex topic and the presence of charged patches matter. This is even more true at low conductivities for mAbs. So this statement needs to be reworded carefully for mAbs.
Answer: As per the valuable suggestion statement to define pH/pI>1 has been reworded.
2. In the methods section, the authors barely give information on the type of IgG. IgG2? IgG4? What is the pI? How many IgGs were tested? Without this information, this paper is not of much value
Answer: In method section the brief information related to Bevacizumab IgG1 antibody has been added with pI and Moleculat weight information.
3. I see what the authors are trying to accomplish -- and it has value -- in terms of reproducibility and robustness, but how is this different from existing published work in this field? I see no mention of other literature that has worked on similar methods (there are authors who have worked on this and published -- fairly common in industry)
Answer: This work is different from other published methods in terms of method qualification, sensitivity and time saving experiments. Also this method can be used directly for mass spectrometry characterization as all the materials used for the analysis is mass compatible. For Bevacizumab molecule this method is specific in terms of pH based elution of acidic and basic variants. Also for Bio pharma industry this method is cost effective than salt based time consuming method which also impact HPLC system and column shelf life with longer run time.
Reviewer 2 Report
In general the topic is interesting. the provided results are promising But this manuscript is poorly written. It is missing the right formatting and style in many sections of the manuscript. The references section needs a lot of work. It is too short and out of dating . The introduction section is very poor and it does not provide enough background about the topic. The author is strongly advised to rewrite the whole manuscript according to the journal style and format.
Author Response
In general the topic is interesting. The provided results are promising but this manuscript is poorly written. It is missing the right formatting and style in many sections of the manuscript. The references section needs a lot of work. It is too short and out of dating. The introduction section is very poor and it does not provide enough background about the topic. The author is strongly advised to rewrite the whole manuscript according to the journal style and format.
Answer: Thank you so much for providing valuable feedback. We have rewritten the poorly worded manuscript paragraph and tried to write more focus scientific language. Reference section has been improvised. In introduction section more scientific approach has been added to justify the hypothesis. Background of topic has been elaborated by more references with right formatting and style.
Reviewer 3 Report
Attached

Author Response
- In the article authors use only 8 references. Only 4 of them are published in the past 5 years.
- First, authors should cite original sources for all the past scientific findings that they discuss
In the introduction. For example, authors cannot discuss line 58-65 without any reference.
Similarly, please fix issue for the entire introduction.
- Authors use different citation styles for their references. Authors should carefully cite the
Literature and should be consistent.
Answer: updated references in the manuscript has been done with correct citation. Introduction part reworded with proper references. Citation styles corrected in the document.
- Materials and methods section should be elaborately written as paragraphs including and
Explaining all the materials used and the way you carried out the experiments. It is very difficult
To follow the methods.
Answer: Materials and method section elaboration with scientific approach step by step has been written paragraph wise.
- All the figure titles and table titles should be very descriptive to understand what authors are
Presenting. Please revise ALL the figure titles and table titles
Answer: All the figures and tables description has been explained in details with legends.
- It is very difficult to understand what authors are presenting in the table 01, what does this time
mean and what is the column “Curve” represent?
Answer: Curve represent the gradient mixing ratio. This varies from system to system. For waters UPLC
It is fixed as 6. Explanation has been added in table 01.
- It is only in the discussion section that the authors state they used “bevacizumab as an example”.
This information should be clearly explained in the introduction.
Answer: In introduction and materials/methods part, detailed explanation of Bevacizumab Igg1 antibody is provided.
- Please move the figure 01 to the results section and please be very descriptive when writing the
figure captions. Figure 01 is missing the x-axis label and axis labels are too small and hard to read.
Answer: Done
- Line 145-162 shouldn’t certainly be in the materials and methods section.
Answer: Done
- In line 168-169 authors mention concentrations, please clarify if these are bevacizumab concentrations.
Answer: Yes it is Bevacizumab IgG1 antibody having actual concentration of 25 mg/mL. For the analysis it is diluted to 3.0 mg/mL in Milli-Q. This information is added as per suggestion.
- In the main text, when explaining research outcomes which are presented in a figure, table or a
graph please indicate the figure number in parenthesis, so that readers know which figure authors
are referring. (eg. For line 171 (figure 3)
Answer: Done
- Readers cannot follow the figure legend of the figure 02. Please provide a clear figure legend
explaining what these curves are and provide very descriptive label.
Answer: Updated
- From line 198-209 authors define the terms precision, accuracy, repeatability, and reproducibility in general. I’m doubted the relevance of the general definitions in the result section. Instead, please relate this to your experimental outcomes.
Answer: As we have done method qualification experiments with all the parameters related to ICH Q2R1 guidelines, each parameter in qualification experiments has been defined and proved for method capability part. The statement having terms precision, accuracy, repeatability, and reproducibility is claimed based on individual result for qualification experiment and it is defined with either % RSD value or recovery calculation for individual parameter.
- Authors’ major claim of the article is developing a pH based CEX UPLC analyzing charge related
variants mAbs but authors didn’t support their claim or justified their results by analyzing or separating the charge variants of the mAbs.
Answer: As the result shows that acidic and basic variants are well separated in 20 min chromatography method without any sample preparation of native antibody, whatever is claimed in hypothesis has been proven after experimental data compilation. Also the method is sensitive to detect charge variants with LOQ is 5.0% of loading amount i.e. 0.15 mg/mL of 100% amount which is 3.0 mg/mL with recovery between 90-110%.
Minor points,
Line 01 should be “post-translational modifications (PTMs)”
Answer: Corrected
There are unnecessary capitalization throughout the article,
Line 19, Charge Variant, Line 62, Charge variants, Line 69 (Cation/Anion), Line 91, Mass, Line 146, Quality, Line 148, Qualification, Line 152 Ion Exchange, Line 176, Mab & Main Peak, Line 178, Main, Line 194 Main Peak, Line 283, Drug, Drug Product.
Answer: Corrected
Line 35, please define CEX here
Answer: Defined
Line 44, Authors state “The isoelectric point (pI), where the net charge of the protein is neutral is pH
Dependent.” Which is technically incorrect. The overall charge of protein is pH dependent, but the
Isoelectric point is an inherent property of the protein.
Answer: Sentence reworded with scientific language.
Line 54, should be “critical quality attributes” since you have (CQAs)
Answer: Corrected
Line 84, there is an incomplete unclear sentence, “The leads to more time and expensive.”
Answer: Corrected and elaborated
Line 85, “Sometimes these experimental protocols lead to unwanted modifications in the protein
Product” – Please be clear and explain these unwanted modification and products.
Answer: Modified
Line 99, There is an incomplete sentence, “he method qualification of a stability…..”
Answer: Modified
Line 263, double periods after “PTMs.”
Answer: Corrected
Round 2
Reviewer 1 Report
The authors have addressed my concerns
Author Response
Thank you
Reviewer 2 Report
The author did a great job to improve the manuscript. This manuscript is suitable for publication in its new version.
Author Response
thank you
Reviewer 3 Report
I recommend authors correcting following minor revisions.
Line one, please remove ‘A’; “Post translational modifications (PTMs)”
Figure 01 is still missing the x-axis label
Figure 02 legend is still difficult to follow and blurry. Same in the Figure 04 & 05
Author Response
Line one, please remove ‘A’; “Post translational modifications (PTMs)”
Answer: Removed.
Figure 01 is still missing the x-axis label.
Answer: X-axis label added.
Figure 02 legend is still difficult to follow and blurry. Same in the Figure 04 & 05
Answer: all the above mentioned legend for the figures are now updated with description.